# Artificial Pasture Grazing System Attenuates Lipopolysaccharide-Induced Gut Barrier Dysfunction, Liver Inflammation, and Metabolic Syndrome by Activating ALP-Dependent Keap1-Nrf2 Pathway

**DOI:** 10.3390/ani13223574

**Published:** 2023-11-19

**Authors:** Qasim Ali, Sen Ma, Boshuai Liu, Ahsan Mustafa, Zhichang Wang, Hao Sun, Yalei Cui, Defeng Li, Yinghua Shi

**Affiliations:** 1Department of Animal Nutrition and Feed Science, College of Animal Science and Technology, Henan Agricultural University, Zhengzhou 450002, China; qasimali@henau.edu.cn (Q.A.); ms0321021@163.com (S.M.); boshuailiu1@126.com (B.L.); zcwang@henau.edu.cn (Z.W.); sunhao@henau.edu.cn (H.S.); yaleicui@henau.edu.cn (Y.C.); leadephone@126.com (D.L.); 2Henan Key Laboratory of Innovation and Utilization of Grassland Resources, Zhengzhou 450002, China; 3Henan Herbage Engineering Technology Research Center, Zhengzhou 450002, China; 4Department of Animal Nutrition, Sichuan Agricultural University, Chengdu 611130, China; dr.ahsan.mustafa@gmail.com

**Keywords:** geese, pasture grazing, intestinal alkaline phosphatase, lipopolysaccharides, systemic inflammation, Keap1-Nrf2

## Abstract

**Simple Summary:**

The present study investigated the role of pasture in ameliorating LPS-ROS-induced gut barrier dysfunction and systemic inflammation in geese. In this study, we found that the pasture was effective in influencing alkaline phosphatase, which, on the one hand, dephosphorylates LPS and, on the other hand, regulates Nrf2 signaling pathway-induced antioxidant enzymes in liver tissues. The ALP and Nrf2 signaling pathways altogether lower the diverse impacts of endotoxemia and oxidative stress by averting systemic inflammation in geese.

**Abstract:**

Introduction. Geese can naturally obtain dietary fiber from pasture, which has anti-inflammatory and antioxidant properties. This study aimed to investigate the inhibitory impacts of pasture on ameliorating LPS-ROS-induced gut barrier dysfunction and liver inflammation in geese. **Materials and methods.** The lipopolysaccharides (LPS), alkaline phosphatase (ALP), reactive oxygen species (ROS), tight junction proteins, antioxidant enzymes, immunoglobulins, and metabolic syndrome were determined using ELISA kits. The Kelch-like-ECH-associated protein 1-Nuclear factor erythroid 2-related factor 2 (Keap1-Nrf2) and inflammatory cytokines were determined using the quantitative reverse transcription PCR (RT-qPCR) method. The intestinal morphology was examined using the Hematoxylin and Eosin (H&E) staining method in ileal tissues. **Results.** Pasture significantly influences nutrient absorption (*p* < 0.001) by ameliorating LPS and ROS-facilitated ileal permeability (*p* < 0.05) and systemic inflammation (*p* < 0.01). Herein, the gut permeability was paralleled by liver inflammation, which was significantly mimicked by ALP-dependent Nrf2 (*p* < 0.0001) and antioxidant enzyme activation (*p* < 0.05). Indeed, the correlation analysis of host markers signifies the importance of pasture in augmenting geese’s health and production by averting gut and liver inflammation. **Conclusions.** Our results provide new insight into the mechanism of the pasture-induced ALP-dependent Nrf2 signaling pathway in limiting systemic inflammation in geese.

## 1. Introduction

Poultry’s health appears to be determined by intestinal homeostasis, which is altered by oxidative stress caused by either heat stress or feed stress [1]. Geese, ducks, broilers, quails, layers, pigeons, and turkeys are continuously exposed to lipopolysaccharides (LPS) through a variety of sources, including feed, water, and tiny dust particles, all of which are known to contain small levels of LPS. However, the diverse gram-negative bacterial community found in the intestines is the main natural source of LPS [2]. Gram-negative bacteria such as *Escherichia coli* [3], *Cyanobacteria* [4], and *Bacterodales* [5] produce LPS as a part of their outer membranes. Then, the LPS undergoes ROS activation-induced systemic inflammation in geese [3]. This further caused leaky gut-induced intestinal fluid loss, gut permeability, nutrient absorption, and diarrhea in geese and broilers [6].

In LPS-challenged animals, LPS activates ROS production and inflammatory responses, which are the main disease-causing elements in modern animal life [7]. Therefore, finding new modifiable feeding strategies that may encounter endogenous LPS-induced disease, oxidative stress, and inflammation can be helpful. Different feeding strategies, such as pectin [8], dietary fiber and threonine [9], and phytochemicals [10], have been applied to deal with LPS in different animal models.

The oral administration of intestinal alkaline phosphatase and different dietary fiber sources such as galactooligosaccharides, glucomannon, and ryegrass has been used to stimulate endogenous ALP activation in rats and geese [3,11]. Intestinal ALP is endogenously secreted from the gut epithelium and is said to regulate gut homeostasis. Loss of ALP activity is correlated with enhanced gut permeability, bacterial translocation, and systemic inflammation [12]. Intestinal ALP is believed to play a vital role in detoxifying LPS, flagellin, and CpG-DNA [13]. However, the studies supporting endogenous ALP regulation in geese to detoxify LPS-induced ROS production, gut permeability, nutrient absorption, and liver inflammation need to be understood.

In this situation, discovering a unique component with anti-oxidative and anti-inflammatory characteristics can be helpful. In light of this, ryegrass, a major source of dietary fiber, possesses a variety of intriguing biological features, such as antioxidant, anti-inflammatory, and antimicrobial properties [3,14].

Geese are herbivores, and due to their unique habit of using fresh grass, ryegrass was believed to improve their health and growth performance. Considering the earlier defined beneficial properties of ryegrass, we aimed to discover its defensive impacts against commercial diet (low dietary fiber source)-induced LPS/ROS-facilitated ileal and liver dysfunction. Furthermore, we investigated ryegrass-dependent ALP-targeted Keap1-Nrf2 signaling pathway-induced antioxidant enzymes, anti-inflammation, immunoglobulin activation, growth performance, intestinal organ development, and metabolic profiles in commercial diet-challenged geese.

## 2. Materials and Methods

### 2.1. Study Design and Animals

A total of one hundred and eighty 1-day-old Wanfu geese were bought from a local hatchery. After brooding (25 d), the geese were divided into two groups. The control group was named the in-house feeding group (IHF), and the treatment group was named the artificial pasture grazing group (AGF). Each group consisted of 90 geese, and the geese were separated into six replicas (15 geese per replicate). All geese had unlimited access to fresh water and food. From 25 to 45 days, IHF geese received grower feed, and from 46 to 90 days, finisher feed (Appendix A). The artificial pasture grazing system was developed in the form of a ryegrass supplement (6:00 am to 18:00) and a commercial diet (only once a day at 19:00).

### 2.2. Sample Collection

At the ages of 45 d, 60 d, and 90 d, the geese were slaughtered using the Muslim method. Individual body weight gain and average feed consumption were assessed on a weekly basis. A simple method for measuring the pasture intake was applied [15]. After cleaning the liver and ileal tissues with phosphate-buffered saline, they were transferred to the liquid nitrogen, followed by storage at −80 °C for additional analysis.

### 2.3. Determination of ALP, LPS, and ROS Levels

Shanghai Enzyme Link Biotechnology Co., Ltd. (Shanghai, China) provided the alkaline phosphatase, lipopolysaccharide, and reactive oxygen species kits. Every experimental methodology was carried out in compliance with the manufacturer’s instructions.

### 2.4. Measurement of Gut Permeability

ELISA Kits were used to measure the quantities of ZO-1, occludin, and claudin from ileal tissues in order to assess gut permeability. Shanghai Enzyme Link Biotechnology Co., Ltd. (Shanghai, China) is the source of all the kits, and the manufacturer’s instructions were followed to complete the experimental methods.

### 2.5. Determination of Antioxidant Activity

Using ELISA kits, the enzymes glutathione reductase (GSR) and heme oxygenase 1 (HO-1) in liver tissues were identified (Shanghai Meilian Biotechnology Co., Ltd., Shanghai, China). Following the manufacturer’s instructions, liver tissues were used to measure malondialdehyde (MDA), glutathione peroxidase (GSH-Px), total antioxidant capacity (T-AOC), total superoxide dismutase (T-SOD), and catalase (CAT) using ELISA kits (Nanjing Jiancheng Bioengineering Institute, Nanjing, China).

### 2.6. Determination of Metabolic Profiles

High-density lipoprotein cholesterol (HDL-C), low-density lipoprotein cholesterol (LDL-C), triglycerides (TG), blood urea nitrogen (BUN), and total cholesterol (T-CHO) were assessed using kits from Nanjing Jiancheng Bioengineering Company (Nanjing, China) following the manufacturer’s recommendations.

### 2.7. Staining with Hematoxylin and Eosin (H&E)

After fixing the ileum tissues with 10% paraformaldehyde, they were stained with Harris’ hematoxylin solution for six hours at 60–70 °C. The tissues were then rinsed with tap water until the water was colorless. Next, the tissues were separated twice with 10% acetic acid and 85% ethanol in water for 2 h and 10 h, respectively, and the tissues were rinsed with tap water again. In the bluing step, the tissues were soaked in saturated lithium carbonate solution for 12 h, then rinsed with tap water, and finally stained with eosin Y ethanol solution for 48 h.

### 2.8. RNA Extraction and cDNA Synthesis for RT-qPCR Analysis

Following the manufacturer’s guidelines, 1 mL of Trizol (MagZol reagent) was used to extract the total RNA from liver and ileal tissues (Magen Biotechnology, Guangzhou, China). A Thermo Scientific, Wilmington, NC, USA, NanoDrop 2000 UV-Vis spectrophotometer was used to measure the concentration and purity of total RNA. Then, in accordance with the manufacturer’s recommendations, RNA was reverse transcribed into cDNA using HiScript^®^ III RT SuperMix for qPCR (+gDNA wiper) (Vazyme, Nanjing, China). With the aid of a NanoDrop 2000 UV-Vis spectrophotometer (Thermo Scientific, Wilmington, NC, USA), the content and purity of total cDNA were evaluated. After that, the cDNA samples were amplified using the ChamQ Universal SYBR qPCR Master Mix from Vazyme Bio-technology (Nanjing, China). Primer3Web version 4.1.0 was used to create gene-specific primers for every gene (Appendix A). The PCR was established on a C1000 Touch PCR Thermal Cycler (BIO-RAD Laboratories, Shanghai, China), and the settings were 40 cycles of 95 °C for 15 s and 60 °C for 30 s. Each measurement was performed three times. Using the 2^−ΔΔCT^ method, the expression levels of mRNA for each of the genes were determined in reliance on beta-actin.

### 2.9. Statistical Analysis

Raw data are presented as mean ± SD. Statistical analysis was performed using SPSS 20.0 software (=D3 SPSS, Inc., 2009, Chicago, IL, USA, www.spss.com, accessed on 8 November 2023). The student’s *t*-test was applied to differentiate the two groups, and *p* < 0.05 was considered statistically significant. Origin 2021 (USA, Northampton) was used to make graphs, and a Pearson correlation analysis was performed with the OECloud tool (https://cloud.oebiotech.cn, accessed on 8 November 2023) to evaluate the relationship between the host markers.

## 3. Results

### 3.1. A Commercial Diet-Dependent Decline in ALP Activity Caused Gut Permeability and Systemic Inflammation

To examine whether ALP plays an important role in geese’s health, we determined the protein level of ALP from ileum tissues. A significant decline in ALP level was shown in pasture-lacking geese from 45 d to 90 d (Figure 1A). As ALP is considered to be closely linked to regulating gut barrier functions, we determined the markers that are usually involved in deteriorating them. The increased concentration of ALP agonists such as LPS and ROS was associated with low dietary fiber source (commercial diet) intake in geese with age (Figure 1B,C). Furthermore, the tight junction proteins for commercial diet-fed geese and artificial pasture-grazing geese were measured from the ileum tissues. Again, the decline in ALP was observed to be involved in reducing ZO-1, occludin, and claudin in pasture-deficient geese (Figure 1D–F). Intestinal hyperpermeability is considered to augment systemic inflammation. Thus, the pro-inflammatory cytokines were measured from the ileal tissues of geese. The systemic inflammatory-related markers were increased in 90-day-old geese compared with younger geese. Herein, the pasture-supplementing geese showed lower mRNA levels of these markers than those of pasture-deficient geese (Figure 1G–K).

### 3.2. Commercial Diet Caused Deterioration of Nutrient Absorption

After confirming ALP-deficient ileal dysbiosis in IHF geese, we sought to investigate intestinal morphology with H&E staining to determine nutrient absorption in ileal tissues. The effect of two feeding systems on the histomorphology of ileal tissues is described in Figure 2 and Table 1. The greater villus height, villus width, villus surface area, and distance between villi were observed in pasture-intake geese compared with commercial diet-feeding geese at 45 d, 60 d, and 90 d. The crypt depth and villus height to crypt depth ratio were improved in IHF geese despite in AGF geese. Furthermore, we determined the membrane thickness of ileal tissues in all sections (Figure 3A,B). The relative thickness of the muscular tonic was higher in AGF geese than that of IHF geese (Table 2). Similarly, the relative thickness of the muscularis mucosa of AGF geese was greater than that of IHF geese from 45 to 90 days (Figure 3B). These results suggest that pasture supplementation was effective in improving nutrient absorption by increasing villus surface area and the overall thickness of the wall of the ileum.

### 3.3. Commercial Diet-Dependent Gut Permeability Is Paralleled by Liver Inflammation

To find out the effects of diet-related IAP alterations in geese, we compared the mRNA levels of inflammatory markers from two feeding groups of geese. For liver tissues, qRT-PCR was applied. The mRNA levels of pro-inflammatory mediators such as *iNOS* and *COX2* were increased in pasture-lacking geese compared with those of pasture-supplementing geese (Figure 4A,B). Furthermore, we determined the mRNA levels of inflammatory cytokines, including *TNF-α*, *IL-6*, and *IL-1β* from liver tissues (Figure 4C–E). Surprisingly, we found the pasture to be effective in decreasing them in geese.

### 3.4. Long-Term Pasture Intake Causes Redox Signaling Pathway Activation

In our previous research work, we proved that high dietary fiber source-dependent ROS production was beneficial in activating the Nrf2 signaling pathway by dissociating Nrf2 from Keap1 in the cecal tissues of meat geese [3]. Herein, we chose the liver tissues and hypothesized that pasture supplementation could regulate redox signaling Keap1-Nrf2 pathway-dependent antibodies to impede liver oxidation. In our study, the mRNA expression levels of *Nrf2* were increased in pasture-supplementing geese instead of *Keap1* (Figure 5A,B). Further, we examined the antioxidants such as HO-1, GSH-PX, GSR, CAT, T-SOD, and T-AOC, which were increased in response to *Nrf2* activation in AGF geese at 45 d, 60 d, and 90 d (Figure 5C–H). To determine whether these antioxidants could mimic the oxidation in the liver of geese, we measured the ROS mediator, such as MDA, from liver tissues. Surprisingly, the protein level of MDA was decreased by pasture supplementation in geese (Figure 5I). It has been described that the regulation of Nrf2 can cause changes in apoptosis and autoantibody production [16]. Therefore, we hypothesized that the activation of the Nrf2 signaling pathway could induce IgA, IgG, and IgM production in response to oxidative stress in live geese. Again, we found the pasture supplementation to be involved in augmenting these antibodies in geese compared with pasture-lacking geese at 45 d, 60 d, and 90 d (Figure 5J–L).

### 3.5. Long-Term Pasture Intake Caused Improved Growth Performance, Intestinal Organ Development, and the Metabolic Profile of Geese

The growth performance of the geese is shown in Figure 6A,B and Table 3. Geese (*n* = 190) at 25 days of age were randomly distributed to groups as follows: in-house feeding geese (IHF) and artificial pasture grazing geese (AGF). Assessment of geese growth performance during the grower phase (45 d) and finisher phase (60 d and 90 d) with unpaired Student T-test (Figure 6 and Table 3) showed that there was a significant difference in average daily feed intake (ADFI) (*p* = 1.62 × 10^−20^, 7.42 × 10^−23^, and 2.61 × 10^−26^) and average daily gain (ADG) (*p* = 1.33 × 10^−14^, 5.60 × 10^−16^, and 4.99 × 10^−22^, respectively). Moreover, the estimated pasture intake (66.09 ± 1.03, 67.98 ± 1.75, and 93.47 ± 1.23 g DM/d, for 45 d, 60 d, and 90 d, respectively) was observed to be helpful in increasing ADG during the finisher phase at 90 d (82.3 ± 0.09 g/d). Briefly, average body weight (ABW) and feed conversion ratio (FCR) were significantly improved during the grower phase (*p* = 2.48 × 10^−6^ and 5.73 × 10^−21^) and the finisher phase (*p* = 1.44 × 10^−7^ and 0.000411 and 1.43 × 10^−17^ and 1.11 × 10^−16^, respectively).

The relative lengths of different sections of the intestine at different growth stages, such as 45 d, 60 d, and 90 d, are represented in Table 4. The relative lengths of the rectum, cecum, ileum, jejunum, duodenum, large intestine, small intestine, and total intestine of pasture-supplementing geese were significantly greater than those of commercial diet-fed geese. Furthermore, the artificial pasture grazing system was seen to be effective in averting metabolic syndrome in geese with a significantly increased lipid profile of the liver and in reducing urea nitrogen levels (Figure 7).

### 3.6. Integrated Analysis of Host Markers

Herein, the findings of our study demonstrate the Pearson correlation of ALP-dependent Keap1-Nrf2 signaling pathway-induced antioxidants with LPS/ROS-induced gut barrier dysfunctions, systemic inflammation, growth performance, and metabolic syndrome in Figure 8. As shown in Figure 8A–C, the ALP, *Nrf2*, HO-1, GSH-PX, GSR, CAT, T-SOD, and T-AOC were significantly positively correlated with HDL-C, tight junction proteins (ZO-1, occludin, and claudin), immunoglobulins (IgA, IgG, and IgM), the relative lengths of the rectum, cecum, ileum, jejunum, duodenum, large intestine, small intestine, and total intestine, and F:G at 45 d, 60 d, and 90 d of sample collection. LPS, ROS, MDA, and *Keap1* were significantly positively correlated with inflammatory cytokines (*IL-1β*, *COX2*, *TNF-α*, *IL-6*, and *iNOS*), metabolic syndrome (LDL-C, T-CHO, TG, and BUN), ABW, and ADG at 45 d, 60 d, and 90 d of sample collection. In summary, the correlation analysis showed that the pasture intake significantly induced ALP and Nrf2-dependent antioxidant enzymes (except MDA) that strongly regulate gut barrier functions, immunity, intestinal organ development, HDL-C, and F:G and attenuate endotoxemia (LPS), oxidative stress, metabolic syndrome, and systemic inflammation in geese.

## 4. Discussion

Chronic low-grade inflammation in liver tissues may result from low dietary fiber diet-induced gut bacterial LPS-mediated ROS production [3]. Finding safe and effective ways to prevent the development of chronic low-grade inflammation is immediately required. The modern poultry feed formulation is built on grains with lower dietary fiber content, which may result in improved growth performance with compromised digestive improvements [17]. According to Huyghebaert et al. [18], these advancements are known to be harmful to visceral growth and to increase the risk of low immunity, intestinal infections, and metabolic illnesses. Therefore, we hypothesized that the dietary fiber present in pasture should ameliorate LPS-ROS-induced gut barrier dysfunction and liver inflammation by activating the ALP-dependent Nrf2 redox signaling pathway in geese.

The consumption of a diet high in fat, carbohydrates, calories, and protein may cause LPS-induced ROS synthesis [19,20], which may cause intestinal mucosal impairment and intestinal permeability [21]. It has been suggested that the decline in ALP activity with aging is paralleled by increased gut permeability and systemic inflammation in humans and mice [22]. However, the role of dietary fiber in inducing the ALP-dependent decline in gut permeability needs to be explored in geese. In our study, we first explained the impacts of gut permeability markers such as LPS and ROS in response to artificial pasture grazing (high dietary fiber source) and commercial diet feeding (low dietary fiber source) systems. Then, we hypothesized that the decline in ALP activity (dependent on feeding systems) may augment ileal barrier functions in geese. Herein, the results of our study are inconsistent with those of Ali et al. [3], in which pasture was effective in augmenting ALP activity in geese. It is well known that ALP dephosphorylates the adherence of LPS and ROS [23]. Here, we observed that pasture supplementation not only induces ALP activity but also reduces the harmful effects of LPS and ROS in pasture-supplementing geese [24].

Considering the key role of the intestinal barrier in maintaining the intestinal environment, disturbance of this barrier during low dietary fiber intake is related to a variety of chronic-related diseases, such as IBD, metabolic syndrome, Alzheimer’s disease, and osteoarthritis [25]. However, the exact role of the intestinal barrier during dietary fiber intake has been poorly studied. A few studies in geese and pigs have reported an increase in gut permeability with low dietary fiber intake [9,26]. In tissues from dietary fiber-deficient geese, decreases in ZO-1, occluding, and claudin were found [3]. In in vitro studies, the upregulation of ALP in Caco-2 and T84 cells significantly increased the mRNA levels of ZO-1 and ZO-2 and ameliorated the LPS-induced intestinal permeability [27]. In this study, we hypothesized that pasture supplementation-dependent ALP activity could prevent gut hyperpermeability in geese. Interestingly, the loss of ALP activity in pasture-lacking geese was linked with increased ileal permeability and reduced protein levels of ZO-1, occluding, and claudin, while pasture supplementation resulted in increased ileal barrier functions and reduced endotoxin (LPS) and ROS concentrations in geese at 45 d, 60 d, and 90 d of age. The possibility of restoring a fully operational intestinal epithelial barrier through pasture supplementation is a very appealing and nutritionally therapeutic intervention.

The dietary fibers that enhance the villus height/crypt depth ratio might also enhance the absorptive ability of the small intestinal epithelium and vice versa [28]. In our study, pasture supplementation significantly increased the height and width of the villi as well as their surface area and distance between them in the ileum of geese. However, the crypt depth and villus height-to-crypt-depth ratio were decreased in AGF geese compared with IHF geese. It is said that the short villi may cause reduced absorption because of two circumstances. First, shortening may occur due to the loss of intestinal surface area, and second, the cells that are destroyed might be the mature cells. Nutrient absorption is essential for the osmotic absorption of water, while water absorption might be impaired because of compromised nutrient absorption. Additionally, the enhanced crypt depth is directly connected to enhanced water secretion. This is one of the primary mechanisms through which toxin-synthesizing bacteria such as *Campylobacter jejuni* [29], *Salmonella* [30], *Brachyspira intermedia* [31], and *E. coli* induce hypersecretory diarrhea. Furthermore, some authors have observed that multiple sources of dietary fiber may have a trophic impact on the small intestinal villi. As an example, in rats, the feeding of pectin (25 g/kg) for 14 days, in contrast to chicken, considerably improved the villus height and crypt depth ratio [32].

LPS has been considered to be closely linked to causing inflammation and several acute and chronic diseases [33], whereas ALP has the ability to dephosphorylate the toxic effects of LPS [34]. Pasture supplementation in geese led to significantly increased ALP and has the capacity to decrease endogenous LPS and ROS in ileal tissues. In addition to a decreased amount of LPS and ROS in AGF geese, we found significantly decreased systemic levels of pro-inflammatory cytokines such as *iNOS*, *COX2*, *IL-6*, *IL-1β*, and *TNF-α*. The levels of these cytokines have been correlated with inflammation in the ileum and liver tissues. In addition, these cytokines have been shown to exert harmful impacts on gut permeability by inducing gastrointestinal mucosal abnormalities [35,36]. We confirmed a significant decrease in *iNOS*, *COX2*, *IL-6*, *IL-1β*, and *TNF-α* in AGF geese, and we found them to be involved in ileal permeability and ileal and liver inflammation in geese lacking pasture supplementation.

In response to LPS/ROS-induced ileal permeability and inflammation, an immune response is activated by the redox-signaling Keap1-Nrf2 pathway. Upon ROS-induced oxidative stress, Keap1 dissociates from Nrf2 and activates Nrf2-targeted antioxidant enzymes [37]. Following the above studies, the ileal ALP-dependent *Nrf2* and its targeted enzymes, such as HO-1, GSH-PX, GSR, CAT, T-SOD, and T-AOC, were augmented in pasture supplementing geese at 45 d, 60 d, and 90 d. Malondialdehyde (MDA) is considered a weapon for oxidative stress that causes liver disorders and generates liver fibrosis, non-alcoholic fatty liver disease, hepatitis C, etc., [38]. Furthermore, MDA deteriorated the quality of broiler meat by destroying muscle mitochondrial protein and lipids [39]. Luckily, we found that the pasture supplementation was effective in reducing DMA levels in liver tissues, thus protecting the geese from fatty liver diseases. It is said that the endogenous regulation of Nrf2-targeted enzymes can stimulate adaptive immunity through immunoglobulin production [16]. Herein, with increasing antioxidant enzyme levels, the IgA, IgG, and IgM protein levels were enhanced in pasture-supplementing geese, suggesting that Nrf2 may be involved in regulating adaptive immunity in geese.

Geese like to eat grass, and due to their distinctive behavior of using high-fiber feeds, ryegrass was considered to promote growth performance, intestinal organ development, and the metabolic profile of the liver. The chemical composition of feed was differentiated on the basis of crude protein (%), crude fat (%), ash (%), dry matter (%), moisture (%), neutral detergent fiber (%), acid detergent fiber (%), calcium (%), and phosphorous (%). The overall feed intake (commercial diet plus grass (g DM/d)) of AGF geese was significantly higher compared with IHF geese. However, the average body weight of IHF geese was significantly greater than that of AGF geese at 45 d, 60 d, and 90 d. It is interesting to note that the average daily gain (g/d) of AGF geese was significantly improved at 90 d compared with IHF geese. This might be due to the pasture intake that impedes the harmful impacts of oxidative stress (i.e., caused by heat stress or feed stress) in geese. Furthermore, the feed-to-gain ratio (or feed conversion ratio) was significantly improved in AGF geese compared with those of IHF geese at 45 d, 60 d, and 90 d. Following the reports of Ling et al. [40] and J. Chen et al. [41], the dietary fiber present in the ryegrass was sufficient to improve ADFI, ADG, and F:G in geese.

Intestinal organ development describes nutrient digestibility and might be dependent on the F:G of the whole body. Therefore, in this study, the development of intestinal organs was measured, which included the rectum, cecum, ileum, jejunum, duodenum, large intestine, small intestine, and total intestine. In the current study, pasture supplementation had significant effects on the relative lengths of the above-mentioned intestinal organs, which were supported by reports that a high dietary fiber diet mainly modified the relative lengths of the intestines of adult genders, geese, and broilers [40,42,43].

Liver biochemical indexes are considered important in determining the metabolism and health status of geese. The concentrations of HDL-C, LDL-C, and T-CHO reflect the levels of lipid metabolism. In a few studies, the 11% accumulation of crude fiber showed improved blood lipid metabolism in broilers [40]. In the present study, a sufficient amount of pasture supplementation was effective in reducing T-CHO and LDL-C and inducing HDL-C in the liver tissues of geese. Additionally, triglycerides are stored in the liver, where they are confined in cytoplasmic lipid droplets. Its higher quantity in the liver causes NAFLD, dyslipidemia, obesity, and type 2 diabetes [44]. Fortunately, our results were different from those of studies in which pasture supplementation restricted the higher accumulation of TG in the liver of geese. Urea nitrogen is a waste product of protein that is taken in the form of food. It is stored in the liver and then travels through the blood to the kidneys, where it is then filtered out of the blood. Urea nitrogen describes how well the kidneys are working. Few studies regarding dietary fiber describe the functions of blood urea nitrogen [45]. Following the previous studies, the dietary fiber present in the pasture was sufficient to control the working of the kidneys by ameliorating excess BUN production in the liver of geese [46].

Lastly, we described the Pearson correlation analysis of ALP-dependent Keap1-Nrf2 signaling pathway-induced antioxidants with LPS/ROS-induced gut barrier dysfunctions, systemic inflammation, growth performance, and metabolic syndrome. The results of this correlation analysis showed that the pasture intake significantly induced ALP and Nrf2-dependent antioxidant enzymes (except MDA) that strongly regulate gut barrier functions, immunity, intestinal organ development, HDL-C, and F:G and attenuate endotoxemia (LPS), oxidative stress, metabolic syndrome, and systemic inflammation in geese.

## 5. Conclusions

In conclusion, artificial pasture grazing system-induced alkaline phosphatase production seems to restore intestinal health by targeting LPS/ROS-induced intestinal barrier dysfunction, systemic inflammation, nutrient absorption, and metabolic syndrome. The mechanisms behind its action may be related to Keap1-Nrf2 signaling pathway-induced antioxidant enzymes, immunoglobulins, growth performance, and lipid metabolism. The beneficial effects of artificial pasture grazing systems as an alternative source of dietary fiber may help to understand gut barrier function-related diseases in animals.

## Figures and Tables

**Figure 1 animals-13-03574-f001:**
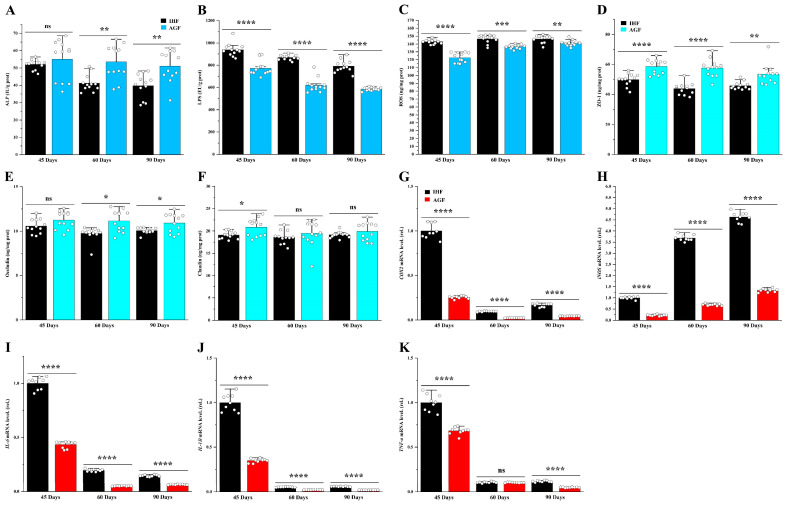
A commercial diet-dependent decline in ALP activity caused gut permeability and systemic inflammation. Protein levels of (**A**) ileal ALP, (**B**) ileal LPS, and (**C**) ileal ROS; Ileal tight junction proteins (**D**) ZO-1, (**E**) occludin, and (**F**) claudin measured by ELISA kits; mRNA levels of (**G**) ileal *COX2*, (**H**) ileal *iNOS*, (**I**) ileal *IL-6*, (**J**) ileal *IL-1β*, and (**K**) ileal *TNF-α*. IHF: in-house feeding system; AGF: artificial pasture grazing system. Data were presented as mean ± SD (*n* = 6). Data with * *p* < 0.05, ** *p* < 0.01, *** *p* < 0.001, **** *p* < 0.0001 were significant; ns: not significant (Student’s *t*-test, *p* < 0.05).

**Figure 2 animals-13-03574-f002:**
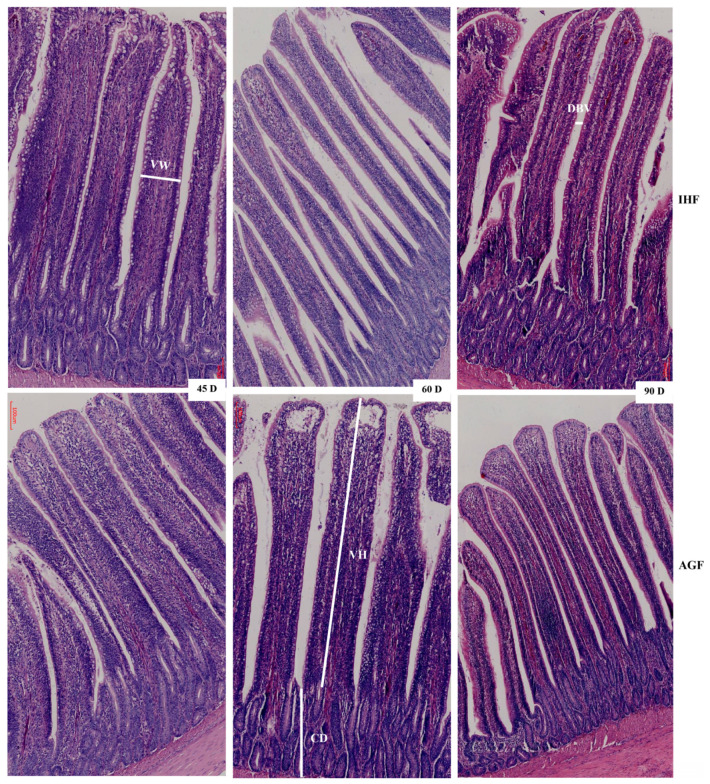
Commercial diet caused deterioration of nutrient absorption. VH: villus height; VW: villus width; DBV: distance between two villi; CD: crypt depth; IHF: in-house feeding system; AGF: artificial pasture grazing system.

**Figure 3 animals-13-03574-f003:**
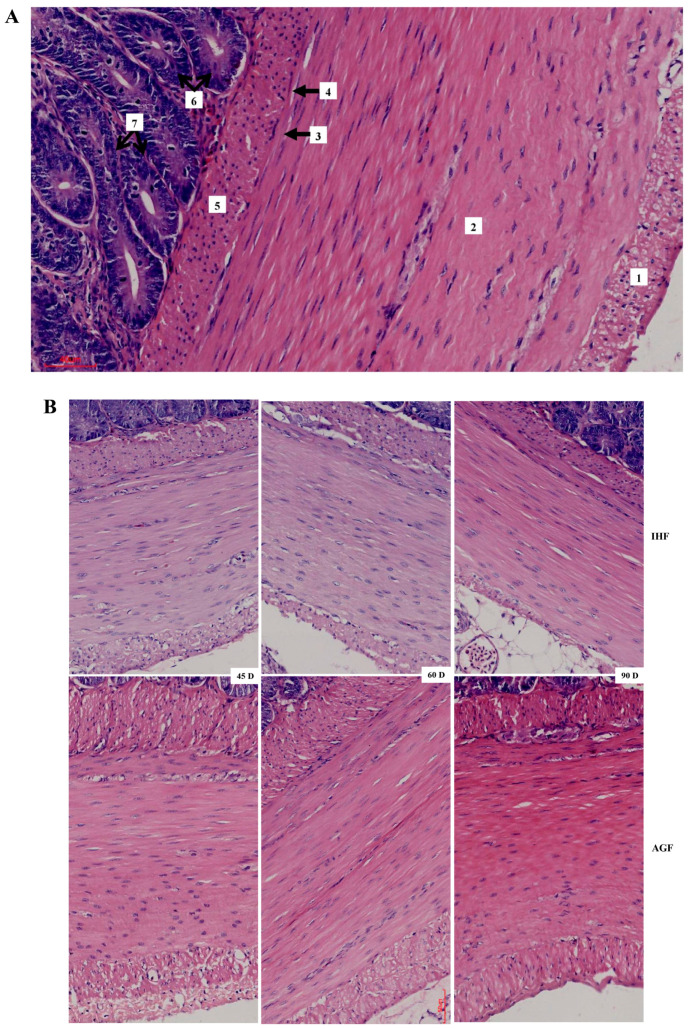
Commercial diet caused deterioration of nutrient absorption by affecting the thickness of ileal muscular tonic and muscularis mucosa (50 µm). (**A**) Light micrograph of the wall of ileum tissues (hematoxylin and eosin). 1: outer layer of muscular tonic; 2: inner layer of muscular tonic; 3: outer layer of lamina muscularis mucosa; 4: ganglion of submucosal nerve plexus; 5: inner layer of lamina muscularis mucosa; 6: crypts; and 7: pillars of unstriated muscle cells (between crypts). (**B**) Comparison of the cecal membrane thickness of geese with different feeding systems (50 µm).

**Figure 4 animals-13-03574-f004:**
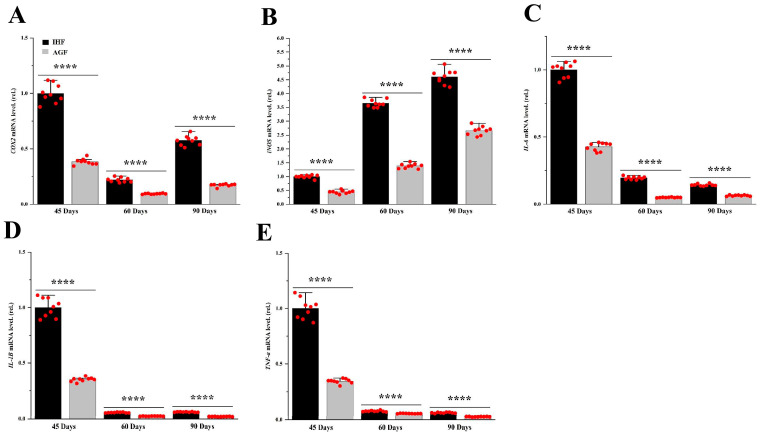
Commercial diet-dependent gut permeability is paralleled by liver inflammation. mRNA levels of (**A**) liver *COX2*, (**B**) liver *iNOS*, (**C**) liver *IL-6*, (**D**) liver *IL-1β*, and (**E**) liver *TNF-α* between the two feeding groups. Data were presented as mean ± SD (*n* = 6). Data with **** *p* < 0.0001 were significant (Student’s *t*-test, *p* < 0.05).

**Figure 5 animals-13-03574-f005:**
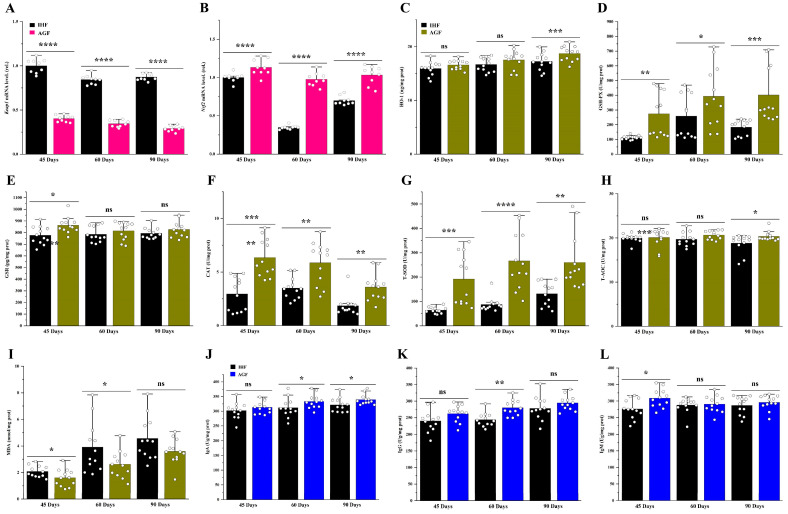
Long-term pasture intake caused redox signaling pathway activation. mRNA levels of (**A**) liver *Keap1* and (**B**) liver *Nrf2*. (**C**–**L**) Protein levels of (**C**) liver HO-1, (**D**) liver GSH-Px, (**E**) liver GSR, (**F**) liver CAT, (**G**) liver T-SOD, (**H**) liver T-AOC, (**I**) liver MDA, (**J**) liver IgA, (**K**) liver IgG, and (**L**) liver IgM between the two feeding groups. Data were presented as mean ± SD (*n* = 6). Data with * *p* < 0.05, ** *p* < 0.01, *** *p* < 0.001, **** *p* < 0.0001 were significant; ns: not significant (Student’s *t*-test, *p* < 0.05).

**Figure 6 animals-13-03574-f006:**
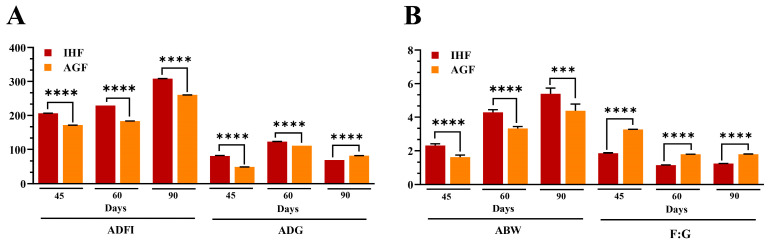
Long-term high dietary fiber diet caused improved growth performance of geese. (**A**) ADFI and ADG measurement on 45 d, 60 d, and 90 d, and (**B**) ABW and FCR measurement on 45 d, 60 d, and 90 d from the two feeding groups. Data were presented as mean ± SD (*n* = 6). Data with *** *p* < 0.001, **** *p* < 0.0001 were significant (Student’s *t*-test, *p* < 0.05).

**Figure 7 animals-13-03574-f007:**
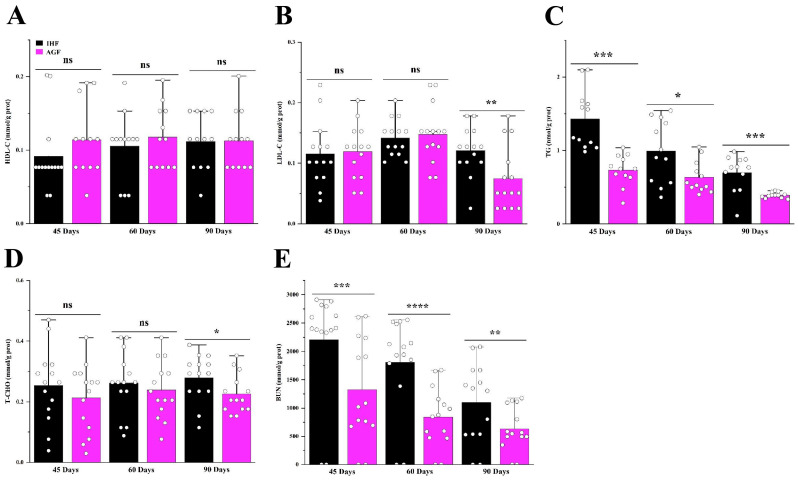
Long-term high dietary fiber diet caused improved metabolic profile of geese. Protein levels of (**A**) liver HDL-C, (**B**) liver LDL-C, (**C**) liver TG, (**D**) liver T-CHO, and (**E**) liver BUN between the two feeding groups. Data were presented as mean ± SD (*n* = 6). Data with * *p* < 0.05, ** *p* < 0.01, *** *p* < 0.001, **** *p* < 0.0001 were significant; ns: not significant (Student’s *t*-test, *p* < 0.05).

**Figure 8 animals-13-03574-f008:**
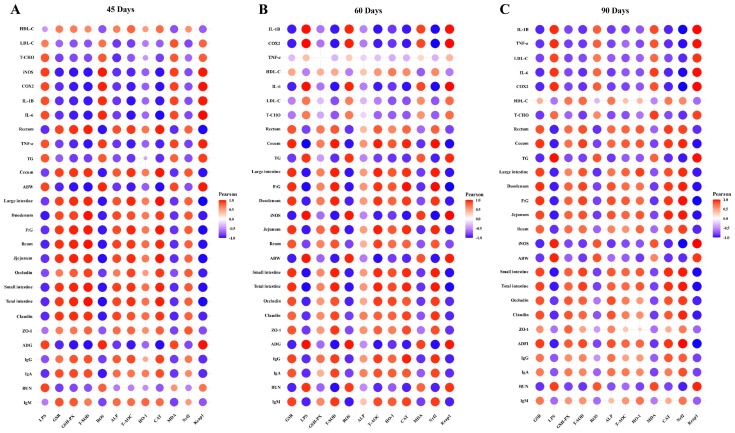
Pearson correlation analysis of ALP-dependent Keap1-Nrf2 signaling pathway-induced antioxidants with LPS/ROS-induced gut barrier dysfunctions, systemic inflammation, growth performance, and metabolic syndrome at 45 d (**A**), 60 d (**B**), and (**C**) 90 d. In the correlation heatmaps, the significant correlations were represented as (*r* > 0.52 or *r* < −0.52, *p* < 0.01). The size of the circle and its color intensity are proportional to the correlation values.

**Table 1 animals-13-03574-t001:** Commercial diet caused deterioration of nutrient absorption. Data expressed as mean ± SD (*n* = 6).

Parameters	45 Days			60 Days			90 Days		
	IHF	AGF	*p* > 0.05	IHF	AGF	*p* > 0.05	IHF	AGF	*p* > 0.05
Inner layer (µm)	51.82 ± 5.76	59.5 ± 6.13	0.020	54.51 ± 5.77	80 ± 8.62	0.000	48.6 ± 7.3	70.29 ± 5.69	0.000
Outer layer (µm)	10.32 ± 2.22	14.36 ± 3.51	0.020	10.97 ± 2.25	19.25 ± 3.63	0.000	8.53 ± 1.18	15.48 ± 2.13	0.000
Total (µm)	62.14 ± 5	73.86 ± 9.04	0.010	65.48 ± 3.93	99.26 ± 9.26	0.000	57.13 ± 7.57	85.77 ± 6.8	0.000
Relative thickness of muscular tonic (µm)	26.98 ± 3.13	45.6 ± 6.55	0.000	15.34 ± 1.33	29.81 ± 2.44	0.000	10.69 ± 1.91	19.76 ± 2.71	0.000
Inner layer (µm)	8.89 ± 1.82	14.25 ± 3.66	0.005	8.5 ± 1.11	21.37 ± 3.68	0.000	6.53 ± 1.27	11.68 ± 3.4	0.003
Outer layer (µm)	1.07 ± 0.17	1.58 ± 0.21	0.000	1.16 ± 0.25	1.61 ± 0.31	0.010	0.99 ± 0.15	1.68 ± 0.33	0.000
Total (µm)	9.95 ± 1.94	15.83 ± 3.49	0.002	9.65 ± 1.01	22.97 ± 3.64	0.000	7.52 ± 1.32	13.35 ± 3.48	0.002
Relative thickness of muscularis mucosa (µm)	4.33 ± 0.99	9.76 ± 2.21	0.000	2.25 ± 0.19	6.89 ± 0.99	0.000	1.41 ± 0.32	3.13 ± 1.05	0.002

**Table 2 animals-13-03574-t002:** Commercial diet caused deterioration of nutrient absorption by affecting the thickness of ileal muscular tonic and muscularis mucosa (50 µm). In-house feeding system (IHF) and artificial pasture grazing system (AGF). Data expressed as mean ± SD (*n* = 6).

Parameters	45 Days			60 Days			90 Days		
	IHF	AGF	*p* > 0.05	IHF	AGF	*p* > 0.05	IHF	AGF	*p* > 0.05
Inner layer (µm)	51.82 ± 5.76	59.5 ± 6.13	0.020	54.51 ± 5.77	80 ± 8.62	0.000	48.6 ± 7.3	70.29 ± 5.69	0.000
Outer layer (µm)	10.32 ± 2.22	14.36 ± 3.51	0.020	10.97 ± 2.25	19.25 ± 3.63	0.000	8.53 ± 1.18	15.48 ± 2.13	0.000
Total (µm)	62.14 ± 5	73.86 ± 9.04	0.010	65.48 ± 3.93	99.26 ± 9.26	0.000	57.13 ± 7.57	85.77 ± 6.8	0.000
Relative thickness of muscular tonic (µm)	26.98 ± 3.13	45.6 ± 6.55	0.000	15.34 ± 1.33	29.81 ± 2.44	0.000	10.69 ± 1.91	19.76 ± 2.71	0.000
Inner layer (µm)	8.89 ± 1.82	14.25 ± 3.66	0.005	8.5 ± 1.11	21.37 ± 3.68	0.000	6.53 ± 1.27	11.68 ± 3.4	0.003
Outer layer (µm)	1.07 ± 0.17	1.58 ± 0.21	0.000	1.16 ± 0.25	1.61 ± 0.31	0.010	0.99 ± 0.15	1.68 ± 0.33	0.000
Total (µm)	9.95 ± 1.94	15.83 ± 3.49	0.002	9.65 ± 1.01	22.97 ± 3.64	0.000	7.52 ± 1.32	13.35 ± 3.48	0.002
Relative thickness of muscularis mucosa (µm)	4.33 ± 0.99	9.76 ± 2.21	0.000	2.25 ± 0.19	6.89 ± 0.99	0.000	1.41 ± 0.32	3.13 ± 1.05	0.002

**Table 3 animals-13-03574-t003:** Long-term pasture intake caused improved growth performance of geese. Data expressed as mean ± SD (*n* = 6). * AGF, the average grass intake (g DM/d) for 45, 60, and 90 days was 66.09 ± 1.03, 67.98 ± 1.75, and 93.47 ± 1.23, respectively. The *p*-value ** is determined on the basis of the commercial diet fed to both the experimental groups (excluding grass intake).

Age, d	Parameters	IHF	* AGF	** *p*-Value
45 d	ADFI (g/d)	206.3 ± 0.27	171.37 ± 0.23	0.000
60 d		229.21 ± 0.01	183.3 ± 0.27	0.000
90 d		308.3 ± 0.09	260.4 ± 0.09	0.000
45 d	ABW (kg)	2.31 ± 0.13	1.63 ± 0.14	0.000
60 d		4.28 ± 0.16	3.33 ± 0.11	0.000
90 d		5.39 ± 0.34	4.38 ± 0.4	0.000
45 d	ADG (g/d)	81.5 ± 1.26	49.06 ± 0.12	0.000
60 d		123.15 ± 0.35	110.88 ± 0.02	0.000
90 d		69.38 ± 0.02	82.3 ± 0.09	0.000
45 d	F:G	1.87 ± 0.01	3.27 ± 0.01	0.000
60 d		1.16 ± 0.01	1.8 ± 0.01	0.000
90 d		1.25 ± 0.01	1.81 ± 0.01	0.000

**Table 4 animals-13-03574-t004:** Long-term pasture intake caused improved intestinal organ development of geese (cm/Kg. Data expressed as mean ± SD (*n* = 6).

Age, d	Parameters	IHF	AGF	*p*-Value
45 d	Rectum	7.04 ± 0.6	8.9 ± 0.84	0.001
60 d		3.94 ± 0.53	5.22 ± 0.46	0.001
90 d		2.99 ± 0.3	3.99 ± 0.37	0.000
45 d	Cecum	10.02 ± 0.63	13.87 ± 1.1	0.001
60 d		5.08 ± 0.36	7.93 ± 0.35	0.000
90 d		3.94 ± 0.17	6.19 ± 0.3	0.000
45 d	Ileum	39.9 ± 4.16	47.78 ± 4.24	0.004
60 d		23.12 ± 1.43	29.57 ± 0.56	0.000
90 d		16.42 ± 1	24.31 ± 1.75	0.000
45 d	Jejunum	42.13 ± 3.18	49.34 ± 8.1	0.030
60 d		23.2 ± 0.73	29.49 ± 1.7	0.000
90 d		17.23 ± 1.41	22.95 ± 1.13	0.000
45 d	Duodenum	20.04 ± 1.1	27.26 ± 2.11	0.000
60 d		13.98 ± 2.42	16.29 ± 1.21	0.030
90 d		9.36 ± 1.71	12.18 ± 1.77	0.010
45 d	Large intestine	17.06 ± 0.9	22.77 ± 1.69	0.000
60 d		9.02 ± 0.67	13.15 ± 0.68	0.000
90 d		6.94 ± 0.44	10.18 ± 0.58	0.000
45 d	Small intestine	102.07 ± 5.98	124.38 ± 12.34	0.001
60 d		60.31 ± 3.55	75.35 ± 1.99	0.000
90 d		43.01 ± 1.44	59.43 ± 3.06	0.000
45 d	Total intestine	119.13 ± 6.26	147.15 ± 13.93	0.001
60 d		69.32 ± 4.12	88.5 ± 2.41	0.000
90 d		49.94 ± 1.74	69.62 ± 3.51	0.000

## Data Availability

All of the data generated or analyzed during this study are included in this published article.

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
