# Peer review of "Artificial Pasture Grazing System Attenuates Lipopolysaccharide-Induced Gut Barrier Dysfunction, Liver Inflammation, and Metabolic Syndrome by Activating ALP-Dependent Keap1-Nrf2 Pathway"

_animals, 2023, doi:10.3390/ani13223574_

Round 1
Reviewer 1 Report
Comments and Suggestions for Authors
Nice work. Please consider the following:
1. Please add a simple summary before the abstract
2. In the abstract, please make sure to include P-values
3. In the abstract please make sure to include the following sections: introduction, importance of your research, materials and methods, results, and conclusions. Each can be 1-3 sentences in the abstract
4. For RNA please add a table of primers used
5. In line 156 take out the link and find a better way to cite
6. Find a better way to present clearer graphs they seem to be smashed together
7. In the result tables, please make sure to follow the format in terms of page margins. Tables are too long occupying all the width of the page
Comments on the Quality of English LanguagePlease revise general sentences in the document. If necessary, ask for help to a native English speaker
Author Response
Q1. Please add a simple summary before the abstract
Response 1: We have added the summary as per request.
Q2. In the abstract, please make sure to include P-values
Response 2: We have added the P-values as per request.
Q3. In the abstract please make sure to include the following sections: introduction, importance of your research, materials and methods, results, and conclusions. Each can be 1-3 sentences in the abstract
Response 3: We have divided the abstract into introduction, materials and methods, results, and conclusions.
Q4. For RNA please add a table of primers used
Response 4: Thank you for the kind suggestion. We have shown the table of primers in Table S2.
Q5. In line 156 take out the link and find a better way to cite
Response 5: We have added the reference as per request.
Q6. Find a better way to present clearer graphs they seem to be smashed together
Response 6: We have clearly presented the graphs with 600DPI as per request.
Q7. In the result tables, please make sure to follow the format in terms of page margins. Tables are too long occupying all the width of the page
Response 7: We have adjusted the tables as per request.
Reviewer 2 Report
Comments and Suggestions for Authors
Artificial pasture grazing system attenuates lipopolysaccharide-induced gut barrier dysfunction, liver inflammation, and metabolic syndrome by activating ALP-dependent Keap1-Nrf2 pathway
Dear Authors,
the manuscript is interesting, because describes positive effect of the artificial pasture grazing system for decreasing some of metabolic disorders in geese, what can be especially useful during the winter season, or in areas where conditions are not proper to effective production of green forage from conventional pastures. There are some important elements to correct in text: Simple summary subsection, table with composition of diet for IHF treatment/group…
Below I add some suggestions helpful during this process:
Line 6-7
No italics needed in case of the last name/surname.
Line 17
The Simple summary must be added first and the Abstract shouldn’t exceed 200 words.
Line 34-35
Space before the paragraph 35 must be removed.
Line 74-245
Lack of the reference no. 15. Perhaps the line 74, because in text in the line 92 is mentioned about 15 geese and I couldn’t find in text no.15 from the line 74, until no. 16 in line 245.
Line 94
Lack of Table S1 with the ingredient and chemical composition of grower and finisher diets for in-house feeding group (IHF) of geese and the chemical composition of ryegrass can be also added.
Line 183
‘Data with different superscript letters are significantly different..’ is added in description of Figure 1, in this case asterisk are used. To use letters, you need to determine significance maximally to 0,01 (A ,B); 0,05 (a, b). In this case better will be asterisk, but appears question: If such accuracy of p-value is necessarily A and B letter could also describe significance with p-value equal i.e. 0.002 or 0.0003 or 0.0000034?
Line 204
In the Table 1 in header for each day is determined ‘…P>0,05…’ must be p-value, and it is describe maximal by three decimals (0.000 is also acceptable). There will be easier to change orientation of the table on page from portrait in word to landscape or present mean and standard deviation in 2 rows of one :
|
Parameters |
45 D |
… |
||
|
|
IHF |
AGF |
p-value |
… |
|
Inner layer |
51.82 ± 5.76 |
59.50 ± 6.13 |
0.020? |
… |
Line 216
Table 2
The same like in the line 204.
Line 226
Figure 4
The same like in the line 183.
Line 252
Figure 5
The same like in the line 183.
Line 277
Figure 6
The same like in the line 183.
Line 283
Please check if standard deviation value for F:G in day 90 is equal 0.
In case of p-value, better will be to left the value maximally with 3 decimals.
Line 298
Table 4
p-value, maximally with 3 decimals, like in case of rectum.
In text of the manuscript is cm\Kg, must be cm\kg.
Line 302
Figure 7.
The same like in the line 183.
Line 479-611
Line spacing should be the same like in previous parts of the text in manuscript.
Journal abbreviations needed, ie.: no. 3: Frontiers of Imunology – Front. Immunol.
The doi links needed for reference no.: 29-32 and 40-43.
Author Response
Q1: Line 6-7. No italics needed in case of the last name/surname.
Response 1: Done as requested.
Q2: Line 17. The Simple summary must be added first and the Abstract shouldn’t exceed 200 words.
Response 2: A short summary and abstract have been mentioned as highlighted in the manuscript.
Q3: Line 34-35. Space before the paragraph 35 must be removed.
Response 3: It has been removed as suggested.
Q4: Line 74-245. Lack of the reference no. 15. Perhaps the line 74, because in text in the line 92 is mentioned about 15 geese and I couldn’t find in text no.15 from the line 74, until no. 16 in line 245.
Response 4: The reference no 15 is mentioned in both text and reference list. Furthermore, there is no error in referencing.
Q5: Line 94. Lack of Table S1 with the ingredient and chemical composition of grower and finisher diets for in-house feeding group (IHF) of geese and the chemical composition of ryegrass can be also added.
Response 5: Thank you for the corrective suggestions. We really appreciate it. As per the suggestion, the chemical composition of grower and finisher diet as well as the ryegrass have been updated.
Q6: Line 183. ‘Data with different superscript letters are significantly different..’ is added in description of Figure 1, in this case asterisk are used. To use letters, you need to determine significance maximally to 0,01 (A ,B); 0,05 (a, b). In this case better will be asterisk, but appears question: If such accuracy of p-value is necessarily A and B letter could also describe significance with p-value equal i.e. 0.002 or 0.0003 or 0.0000034?
Response 6: Thank you for indicating the error. We have shown the significance data with the help of asterisk only in all the figures.
Q7: Line 204. In the Table 1 in header for each day is determined ‘…P>0,05…’ must be p-value, and it is describe maximal by three decimals (0.000 is also acceptable). There will be easier to change orientation of the table on page from portrait in word to landscape or present mean and standard deviation in 2 rows of one :
|
Parameters |
45 D |
… |
||
|
|
IHF |
AGF |
p-value |
… |
|
Inner layer |
51.82 ± 5.76 |
59.50 ± 6.13 |
0.020? |
… |
Response 7: All the p-values are mentioned in the suggested format (0.000). For better visual convenience the table has been kept in landscape format.
Q8: Line 216. Table 2, the same like in the line 204.
Response 8: The same format is used in Table 2.
Q9: Line 226. Figure 4, the same like in the line 183.
Response 9: The same format is used in all the figures.
Q10: Line 252: Figure 5, the same like in the line 183.
Response 10: The same format is used in all the figures.
Q11: Line 277: Figure 6, the same like in the line 183.
Response 11: The same format is used in all the figures.
Q12: Line 283: Please check if standard deviation value for F:G in day 90 is equal 0. In case of p-value, better will be to left the value maximally with 3 decimals.
Response 12: Thank you for indicating the error. We have updated the standard deviation value for F:G in day 90, that is 1.25±0.01.
Q13: Line 298: Table 4, p-value, maximally with 3 decimals, like in case of rectum. In text of the manuscript is cm\Kg, must be cm\kg.
Response 13: Thank you for pointing out the errors which are now corrected.
Q14: Line 302: Figure 7, the same like in the line 183.
Response 14: The same format is used in all the figures.
Q15: Line 479-611. Line spacing should be the same like in previous parts of the text in manuscript. Journal abbreviations needed, ie.: no. 3: Frontiers of Imunology – Front. Immunol. The doi links needed for reference no.: 29-32 and 40-43.
Response 15: We appreciate the suggestions. All the Journal names have been mentioned as per the journal format. The doi link for the respective reference has been added as far as possible.
Round 2
Reviewer 1 Report
Comments and Suggestions for Authors
1. The tables and graphs have not been modified to follow normal page set up. Graphs and tables should not exceed the page margins.
2. Please upload a version that does not have the "mark-up" it is hard to follow up wit the changes wit all the mess shown.
Comments on the Quality of English LanguageNo issues identified
Author Response
Detailed response to reviewers
Dear Editor,
Manuscript Number: animals-2672187 entitled “Artificial Pasture Grazing System Attenuates Lipopolysaccharide-Induced Gut Barrier Dysfunction, Liver Inflammation, and
Metabolic Syndrome by Activating ALP-Dependent Keap1-Nrf2 Pathway” is re-submitted after major revision to the “Animals” for consideration of publication.
We do appreciate the comments from Animals Production Office and considered them carefully. Please do find a revised version of our manuscript. We hope that the revisions in the manuscript and our accompanying responses will be sufficient to make our manuscript suitable for publication in Animals. We believe that our study will be interesting for the readers of Animals and as well as for a broad community of scientists.
Yours sincerely,
Prof. Dr. Yinghua Shi
Author’s response
Note: All the parts of the manuscript has been carefully edited as per your suggestions and all the changes have been highlighted with green color. Please refer to the following technical terms related to company names, kits, etc., which cannot be changed in the manuscript.
Shanghai Enzyme Link Biotechnology Co., Ltd. provided the alkaline phosphatase, lipopolysaccharide, and reactive oxygen species kits (Shanghai, China).
Shanghai Meilian Biotechnology Co., Ltd. Shanghai, China.
Nanjing Jiancheng Bioengineering Institute, Nanjing, Jiangsu, P.R. China.
Following the manufacturer's guidelines, 1 mL of Trizol (MagZol reagent) was used to extract the total RNA from liver and ileal tissues (Magen Biotechnology, Guangzhou, Guangdong, China).
After that, the cDNA samples were amplified using the ChamQ Universal SYBR qPCR Master Mix from Vazyme Bio-technology (Nanjing, Jiangsu, China).
IHF: in-house feeding system; AGF: artificial pasture grazing system.
VH: villus height; VW: villus width; DBV: distance between two villi; CD: crypt depth; IHF: in-house feeding system; AGF: artificial pasture grazing system.
(A) Light micrograph of the wall of ileum tissues (hematoxylin and eosin). 1: outer layer of muscular tonic; 2: inner layer of muscular tonic; 3: outer layer of lamina muscularis mucosa; 4: ganglion of submucosal nerve plexus; 5: inner layer of lamina muscularis mucosa; 6: crypts; and 7: pillars of unstriated muscle cells (between crypts). (B) Comparison of the cecal membrane thickness of geese with different feeding systems (50µm).
Data with *P < 0.05, **P < 0.01, ***P < 0.001, ****P < 0.0001 were significant (Student's t-test, P < 0.05).
Animals Production Office
Reviewer #1:
Q1. Please add a simple summary before the abstract
Response 1: We have added the summary as per request.
Q2. In the abstract, please make sure to include P-values
Response 2: We have added the P-values as per request.
Q3. In the abstract please make sure to include the following sections: introduction, importance of your research, materials and methods, results, and conclusions. Each can be 1-3 sentences in the abstract
Response 3: We have divided the abstract into introduction, materials and methods, results, and conclusions.
Q4. For RNA please add a table of primers used
Response 4: Thank you for the kind suggestion. We have shown the table of primers in Table S2.
Q5. In line 156 take out the link and find a better way to cite
Response 5: We have added the reference as per request.
Q6. Find a better way to present clearer graphs they seem to be smashed together
Response 6: We have clearly presented the graphs with 600DPI as per request.
Q7. In the result tables, please make sure to follow the format in terms of page margins. Tables are too long occupying all the width of the page
Response 7: We have adjusted the tables as per request.
Reviewer 2 Report
Comments and Suggestions for Authors I have no further suggestions, all previous ones have been included in the revised version of the manuscript.
Author Response
Detailed response to reviewers
Dear Editor,
Manuscript Number: animals-2672187 entitled “Artificial Pasture Grazing System Attenuates Lipopolysaccharide-Induced Gut Barrier Dysfunction, Liver Inflammation, and
Metabolic Syndrome by Activating ALP-Dependent Keap1-Nrf2 Pathway” is re-submitted after major revision to the “Animals” for consideration of publication.
We do appreciate the comments from Animals Production Office and considered them carefully. Please do find a revised version of our manuscript. We hope that the revisions in the manuscript and our accompanying responses will be sufficient to make our manuscript suitable for publication in Animals. We believe that our study will be interesting for the readers of Animals and as well as for a broad community of scientists.
Yours sincerely,
Prof. Dr. Yinghua Shi
Author’s response
Note: All the parts of the manuscript has been carefully edited as per your suggestions and all the changes have been highlighted with green color. Please refer to the following technical terms related to company names, kits, etc., which cannot be changed in the manuscript.
Shanghai Enzyme Link Biotechnology Co., Ltd. provided the alkaline phosphatase, lipopolysaccharide, and reactive oxygen species kits (Shanghai, China).
Shanghai Meilian Biotechnology Co., Ltd. Shanghai, China.
Nanjing Jiancheng Bioengineering Institute, Nanjing, Jiangsu, P.R. China.
Following the manufacturer's guidelines, 1 mL of Trizol (MagZol reagent) was used to extract the total RNA from liver and ileal tissues (Magen Biotechnology, Guangzhou, Guangdong, China).
After that, the cDNA samples were amplified using the ChamQ Universal SYBR qPCR Master Mix from Vazyme Bio-technology (Nanjing, Jiangsu, China).
IHF: in-house feeding system; AGF: artificial pasture grazing system.
VH: villus height; VW: villus width; DBV: distance between two villi; CD: crypt depth; IHF: in-house feeding system; AGF: artificial pasture grazing system.
(A) Light micrograph of the wall of ileum tissues (hematoxylin and eosin). 1: outer layer of muscular tonic; 2: inner layer of muscular tonic; 3: outer layer of lamina muscularis mucosa; 4: ganglion of submucosal nerve plexus; 5: inner layer of lamina muscularis mucosa; 6: crypts; and 7: pillars of unstriated muscle cells (between crypts). (B) Comparison of the cecal membrane thickness of geese with different feeding systems (50µm).
Data with *P < 0.05, **P < 0.01, ***P < 0.001, ****P < 0.0001 were significant (Student's t-test, P < 0.05).
Animals Production Office
Reviewer #2:
Q1: Line 6-7. No italics needed in case of the last name/surname.
Response 1: Done as requested.
Q2: Line 17. The Simple summary must be added first and the Abstract shouldn’t exceed 200 words.
Response 2: A short summary and abstract have been mentioned as highlighted in the manuscript.
Q3: Line 34-35. Space before the paragraph 35 must be removed.
Response 3: It has been removed as suggested.
Q4: Line 74-245. Lack of the reference no. 15. Perhaps the line 74, because in text in the line 92 is mentioned about 15 geese and I couldn’t find in text no.15 from the line 74, until no. 16 in line 245.
Response 4: The reference no 15 is mentioned in both text and reference list. Furthermore, there is no error in referencing.
Q5: Line 94. Lack of Table S1 with the ingredient and chemical composition of grower and finisher diets for in-house feeding group (IHF) of geese and the chemical composition of ryegrass can be also added.
Response 5: Thank you for the corrective suggestions. We really appreciate it. As per the suggestion, the chemical composition of grower and finisher diet as well as the ryegrass have been updated.
Q6: Line 183. ‘Data with different superscript letters are significantly different..’ is added in description of Figure 1, in this case asterisk are used. To use letters, you need to determine significance maximally to 0,01 (A ,B); 0,05 (a, b). In this case better will be asterisk, but appears question: If such accuracy of p-value is necessarily A and B letter could also describe significance with p-value equal i.e. 0.002 or 0.0003 or 0.0000034?
Response 6: Thank you for indicating the error. We have shown the significance data with the help of asterisk only in all the figures.
Q7: Line 204. In the Table 1 in header for each day is determined ‘…P>0,05…’ must be p-value, and it is describe maximal by three decimals (0.000 is also acceptable). There will be easier to change orientation of the table on page from portrait in word to landscape or present mean and standard deviation in 2 rows of one :
|
Parameters |
45 D |
… |
||
|
|
IHF |
AGF |
p-value |
… |
|
Inner layer |
51.82 ± 5.76 |
59.50 ± 6.13 |
0.020? |
… |
Response 7: All the p-values are mentioned in the suggested format (0.000). For better visual convenience the table has been kept in landscape format.
Q8: Line 216. Table 2, the same like in the line 204.
Response 8: The same format is used in Table 2.
Q9: Line 226. Figure 4, the same like in the line 183.
Response 9: The same format is used in all the figures.
Q10: Line 252: Figure 5, the same like in the line 183.
Response 10: The same format is used in all the figures.
Q11: Line 277: Figure 6, the same like in the line 183.
Response 11: The same format is used in all the figures.
Q12: Line 283: Please check if standard deviation value for F:G in day 90 is equal 0. In case of p-value, better will be to left the value maximally with 3 decimals.
Response 12: Thank you for indicating the error. We have updated the standard deviation value for F:G in day 90, that is 1.25±0.01.
Q13: Line 298: Table 4, p-value, maximally with 3 decimals, like in case of rectum. In text of the manuscript is cm\Kg, must be cm\kg.
Response 13: Thank you for pointing out the errors which are now corrected.
Q14: Line 302: Figure 7, the same like in the line 183.
Response 14: The same format is used in all the figures.
Q15: Line 479-611. Line spacing should be the same like in previous parts of the text in manuscript. Journal abbreviations needed, ie.: no. 3: Frontiers of Imunology – Front. Immunol. The doi links needed for reference no.: 29-32 and 40-43.
Response 15: We appreciate the suggestions. All the Journal names have been mentioned as per the journal format. The doi link for the respective reference has been added as far as possible.